# Effect of Melatonin on Chemoresistance Exhibited by Spheres Derived from Canine Mammary Carcinoma Cells

**DOI:** 10.3390/ani14081229

**Published:** 2024-04-19

**Authors:** Dania Cataldo, Guillermo Aravena, Alejandro Escobar, Julio C. Tapia, Oscar A. Peralta, Cristian G. Torres

**Affiliations:** 1Centralized Laboratory of Veterinary Research, Faculty of Animal and Veterinary Sciences, Universidad de Chile, Santiago 8820808, Chile; dania.cataldo2@gmail.com (D.C.); gu.aravena.c@gmail.com (G.A.); 2Laboratory of Biomedicine, Department of Clinical Sciences, Faculty of Animal and Veterinary Sciences, Universidad de Chile, Santiago 8820808, Chile; 3Laboratory of Cell and Molecular Biology, Dental Sciences Research Institute, Faculty of Dentistry, Universidad de Chile, Santiago 8380453, Chile; janodvm@gmail.com; 4Cell and Molecular Biology Program, Biomedical Sciences Institute, Faculty of Medicine, Universidad de Chile, Santiago 8380453, Chile; jtapiapineda@uchile.cl; 5School of Veterinary Medicine, Pontificia Universidad Catolica de Chile, Santiago 7820435, Chile; oscarperalta@uc.cl

**Keywords:** canine mammary cancer, cancer stem cells, melatonin

## Abstract

**Simple Summary:**

Mammary cancer is a frequent disease in female dogs, where a high proportion of cases correspond to malignant tumors that may exhibit drug resistance. There is a cancer cell subpopulation called cancer stem cells (CSCs), capable of forming spheres in vitro and resisting anti-tumor treatments. Melatonin has shown antitumor effects on mammary tumor cells; however, its effects have been poorly evaluated in canine mammary CSCs. This study aimed to analyze the effect of melatonin on the chemoresistance exhibited by spheres derived from canine mammary carcinoma to cytotoxic drugs such as doxorubicin and mitoxantrone. Melatonin reduced viability only in CF41.Mg spheres, without inducing an additive effect when co-incubated with cytotoxic drugs. Moreover, the hormone triggers these effects in a way that does not involve its specific MT1 receptor. In CF41.Mg spheres, the relative gene expression of *ABCG2* and *MDR1*—multidrug resistance molecules—was decreased in response to the hormone. These results indicate that melatonin negatively modulates the cell survival of spheres derived from CF41.Mg cells, in a way that is independent of its MT1 receptor. These effects did not counteract the resistance to doxorubicin and mitoxantrone, even though the hormone negatively regulates the expression of *MDR1* and *ABCG2*.

**Abstract:**

Mammary cancer is a frequent disease in female dogs, where a high proportion of cases correspond to malignant tumors that may exhibit drug resistance. Within the mammary tumor microenvironment, there is a cell subpopulation called cancer stem cells (CSCs), which are capable of forming spheres in vitro and resisting anti-tumor treatments, partly explaining the recurrence of some tumors. Previously, it has been described that spheres derived from canine mammary carcinoma cells CF41.Mg and REM 134 exhibit stemness characteristics. Melatonin has shown anti-tumor effects on mammary tumor cells; however, its effects have been poorly evaluated in canine mammary CSCs. This study aimed to analyze the effect of melatonin on the chemoresistance exhibited by stem-like neoplastic cells derived from canine mammary carcinoma to cytotoxic drugs such as doxorubicin and mitoxantrone. CF41.Mg and REM 134 cells were cultured in high-glucose DMEM supplemented with fetal bovine serum and L-glutamine. The spheres were cultured in ultra-low attachment plates in DMEM/F12 medium without fetal bovine serum and with different growth factors. The CD44^+^/CD24^−/low^ phenotype was analyzed by flow cytometry. The viability of sphere-derived cells (MTS reduction) was studied in the presence of melatonin (0.1 or 1 mM), doxorubicin, mitoxantrone, and luzindole. In addition, the gene (RT-qPCR) of the multidrug resistance bombs *MDR1* and *ABCG2* were analyzed in the presence of melatonin. Both cell types expressed the *MT1* gene, which encodes the melatonin receptor MT1. Melatonin 1 mM does not modify the CD44^+^/CD24^−/low^ phenotype; however, the hormone reduced viability (*p* < 0.0001) only in CF41.Mg spheres, without inducing an additive effect when co-incubated with cytotoxic drugs. These effects were independent of the binding of the hormone to its receptor MT1, since, by pharmacologically inhibiting them, the effect of melatonin was not blocked. In CF41.Mg spheres, the relative gene expression of *ABCG2* and *MDR1* was decreased in response to the hormone (*p* < 0.001). These results indicate that melatonin negatively modulates the cell survival of spheres derived from CF41.Mg cells, in a way that is independent of its MT1 receptor. These effects did not counteract the resistance to doxorubicin and mitoxantrone, even though the hormone negatively regulates the gene expression of *MDR1* and *ABCG2*.

## 1. Introduction

Mammary tumors are a disease that frequently occurs in reproductively intact female dogs [1,2,3]. Approximately 60% of these tumors are malignant, which implies the capacity for local invasion and metastasis [1,4]. More than 90% of malignant tumors correspond to carcinomas, within which three degrees of malignancy are recognized [5,6]. In cases of high-grade carcinomas (grade III), multimodal therapies are used, including chemotherapy [7]. However, affected animals usually develop drug resistance, decreasing treatment efficacy [2]. Cancer stem cells (CSCs) represent a subset of cancer cells of solid mammary tumors that exhibit self-renewal and can express stemness markers such as CD44^+^/CD24^−/low^, aldehyde-dehydrogenase (ALDH), octamer-binding transcription factor 4 (OCT4), and EpCAM (epithelial cell adhesion molecule or CD326), among others. Moreover, these cells show a tumor initiation capacity and can resist the cytotoxic effects of chemotherapy, which may partly explain post-treatment tumor recurrence [8,9,10,11,12]. It has been described that CSCs derived from CF41.Mg and REM134 canine mammary carcinoma cells exhibit an in vitro sphere-forming capacity (cell structures that grow in anchorage-free conditions and the absence of fetal bovine serum) and chemoresistance to different drugs such as doxorubicin [13,14]. In the case of CF41.Mg spheres, they have also exhibited a relative resistance to paclitaxel and simvastatin [14]. CSCs can develop an escape to chemotherapy through various cellular and molecular mechanisms, such as genetic mutations and/or epigenetic changes, drug efflux due to the high expression of multidrug resistance (MDR) genes, and quiescence, among others [13,15,16,17]. MDR1 (ABCB1) and ABCG2 (BCRP) are involved in increasing the efflux of chemotherapy drugs, which leads to a decrease in their intracellular concentration, favoring a resistance to chemotherapy [16,17]. MDR1 is a transmembrane protein that has multiple drug-binding sites, allowing it to pump a wide variety of substrates into the extracellular space, such as anthracyclines. ABCG2, for its part, is the main transporter associated with drug efflux in resistance associated with mammary cancer, where it can transport chemotherapeutic drugs such as mitoxantrone and other anthracyclines [11,13,15,18]. Both proteins are highly expressed in high-grade mammary carcinomas, being positively associated with malignancy and failures in response to pharmacological treatment [15,19]. Melatonin is a chronobiological hormone secreted mainly by the pineal gland and has shown anti-tumor effects in vitro on canine mammary cancer cells dependent or not on estradiol [20]. The mechanisms involved are diverse and include the inhibition of cell viability, proliferation, and induction of apoptosis, modifying some pathogenic pathways linked to drug resistance [21,22,23,24,25]. These effects are mainly mediated by the binding with its specific G-protein-coupled receptors MT1 and MT2, although it can also act independently of its receptors [20,26,27,28]. The impact of this hormone on mammary CSCs has been poorly studied; nevertheless, there is evidence to suggest that the hormone could regulate this type of cells. In this sense, it has been described in canine and human mammary cancer cells that melatonin negatively modulates the expression of OCT4, a key transcription factor in the development of stemness [29]. Moreover, CF41.Mg spheres are more sensitive to melatonin than CF41.Mg-parental cells [30], which suggests that the hormone specifically modulates this subpopulation of tumor cells, potentially downregulating the chemoresistance. Thus, we hypothesized that melatonin reduces drug resistance in canine mammary CSCs through the downregulation of MDR1 and ABCG2. This work aimed to determine the effect of melatonin on the chemoresistance exhibited by spheres derived from two canine mammary carcinoma lines.

## 2. Methods

### 2.1. Cell Culture

The canine mammary carcinoma cell lines CF41.Mg (CRL-6232, ATCC, Manassas, VA, USA) and REM 134 (12122002, Merck KgaA, Darmstadt, Germany) were cultured in high-glucose DMEM (Dulbecco’s modified Eagle’s medium), supplemented with 10% fetal bovine serum (FBS), 2 mM glutamine (Gibco, Life Technologies, Carlsbad, CA, USA), and penicillin/streptomycin (Sartorius, Beit Haemek, Israel). Both cell types are derived from a spontaneous primary tumor and show a fibroblastoid and polygonal morphology, respectively. They were maintained at 37 °C in a humidified atmosphere with 5% CO_2_ incubator and the culture medium was changed every 48 h after washing the cells with sterile phosphate-buffered saline (PBS, pH 7.4). For cell disaggregation, once 80–90% confluence was reached, they were washed with PBS, and then incubated with 2 mL of Trypsin/EDTA (Trypsin 0.25%, EDTA 0.05%, Sartorius, Beit Haemek, Israel) for 4–10 min. The cells were negative for Mycoplasma through a PCR method (Sartorius, Beit Haemek, Israel). The concentration of live cells was evaluated through the exclusion method with 0.4% trypan blue (Gibco, Life Technologies, Carlsbad, CA, USA) and hemocytometry.

To obtain spheroid cells (spheres), the parental cells (adherent CF41.Mg and REM 134) were detached, washed with PBS, and seeded in sphere-culture medium, on ultra-low-adherence plates (Corning, NY, USA) and in the absence of FBS. CF41.Mg sphere medium contained DMEM-F12, supplemented with 2% B27, 4 µg/mL heparin, 5 µg/mL recombinant human insulin (IRH), 10 ng/mL basic fibroblast growth factor (bFGF), 10 ng/mL epidermal growth factor (EGF), 20 U/mL penicillin G, 20 µg/mL streptomycin, and 0.05 µg/mL amphotericin B [14]. For the REM 134 spheres, the medium contained DMEM-F12 plus 20 nM progesterone, 100 μM putrescine, 30 nM sodium selenite, 25 μg/mL transferrin, 20 μg/mL IRH, 10 ng/mL bFGF, 10 ng/mL EGF, and antibiotics [13].

### 2.2. Flow Cytometry

CF41.Mg and REM 134 cells were seeded in ultra-low-adherence 100 mm plates in pertinent sphere-medium, with a density of 2 × 10^6^ cells/plate. After 48 h, cells were incubated with 1 mM melatonin and pertinent controls (blank and control vehicle) for 24 h. Then, cells were mechanically disaggregated, washed, and incubated with specific labeled antibodies against CD44 (APC rat anti-mouse CD44 clone IM7 (559250)), CD24 (PE rat anti-mouse CD24 clone M1/69 (553262)), and pertinent isotypes control (APC rat IgG2b, clone A95-1; PE rat IgG2b clone A95-1) at 4 °C for 45 min, as described by Torres et al., 2015 [14]. The cell populations were analyzed by flow cytometry using a BD FACSCalibur cytometer (BD Biosciences, San Jose, CA, USA). The data were collected and analyzed by Flowjo Software 7.6.1 (Tree Star Software, Ashland, OR, USA) [14]. At least 3 independent experiments were performed.

### 2.3. Cell Viability Assays

CF41.Mg and REM 134 cells were seeded in ultra-low-adhesion 96-well plates, with a density of 5000 cells/well and 10,000 cells/well, respectively, in 100 µL of sphere-medium in quadruplicate. A blank was left in quadruplicate, which only contained the culture medium. A dose–response curve was performed at different concentrations of mitoxantrone (1–1000 nM) in spheres from both cell lines. After that, melatonin (Sigma-Aldrich-M5250, Darmstadt, Germany) was applied in different concentrations (0, 0.1, and 1 mM) in the presence/absence of doxorubicin, mitoxantrone (1 nM), and luzindole (50 μM). In the melatonin plus chemo-drugs experiments, melatonin was added 1 h before the drugs. To the combinatorial luzindole and melatonin, the former was added 1 h before the hormone. At 48 and 72 h of incubation, cell viability was analyzed through the MTS reduction method [3-(4,5-dimethy-2-yl)-5-(3-carboxymethoxyphenyl)-2-(4-sulfophenyl)-2H-tetrazolium, inner salt, MTS]. For this, 20 µL of MTS (CellTiter 96^®^, Promega, Madison, WI, USA) were applied per 100 µL of culture medium and was incubated for three hours at 37 °C in a humidified atmosphere with 5% CO_2_. The resulting optical density (O.D.) was measured in a multiplate reader at 490 nm. Cell viability, referred to as the proportion of living cells after the experiment, was calculated as a relative value to the control, where the average O.D. of the control group was considered to have 100% viability. At least 3 independent experiments were performed.

### 2.4. qPCR

CF41.Mg spheres were cultured in the absence and presence of 0.1 and 1 mM melatonin for 24 h. Total RNA was isolated using the RNAeasy kit (Qiagen, Redwood, CA, USA) following the manufacturer’s instructions. After total RNA quantification, a reverse transcription (RT) reaction was performed using the Brilliant SYBR Green II RT-PCR kit (Agilent Technologies, Santa Clara, CA, USA). The resulting cDNA was used for amplification using primers specific for *MDR1* and *ABCG2*. *β-actin* was used as an endogenous normalization control. Moreover, *MT1* and *MT2* primers were designed by Primers Blast, from the sequence of exon 2 of the gene coding for *MT1* and *MT2* in dogs (*Canis lupus familiaris*) (Table 1). Real-time PCR was performed using a 2× SYBR green PCR master mix (Agilent Technologies, Santa Clara, CA, USA). Relative expression was determined using the ΔΔCT (relative quantification) analysis protocol.

### 2.5. Data Analysis

The Shapiro–Wilk test was used to determine the type of data distribution. At least three independent experiments were performed in the context of each assay. *T*-test, ANOVA–Bonferroni, or Kruskal test was used to evaluate differences between experimental conditions. *p* ≤ 0.05 was considered significant. The data were analyzed using GraphPad Prism version 8.0.1.

## 3. Results

The gene expression of the melatonin-specific receptors MT1 and MT2 was evaluated by qPCR since there is no published information regarding their expression in spheres derived from REM134 and CF41.Mg cells. A consistent expression of the *MT1* gene was observed, while *MT2* was not expressed in both cell types (Figure 1).

One of the main phenotypic characteristics of mammary CSCs is the high expression of CD44 and the low or lack of presence of CD24. Thus, we evaluated whether melatonin could modulate this expression. Spheres derived from both cell types exhibit a high proportion of CD44^+^/CD24^−/low^ (92.2% ± 3.2 in REM134 spheres; 87.4% ± 1.5 in CF41.Mg spheres), which confirms its stem phenotype. Melatonin 1 mM did not modify this phenotype in both REM134 (92.2% ± 3.2 in control vs. 92.8% ± 3.4 in melatonin condition) and CF41.Mg (87.4% ± 1.5 in control vs. 87.3% ± 2.6 in melatonin condition) (*p* > 0.05), as shown in Figure 2.

As already described, spheres derived from REM134 and CF41.Mg canine mammary carcinoma cells exhibit a relative chemoresistance to doxorubicin [13,14], which was corroborated in this study. To evaluate whether melatonin interferes with this chemoresistance, spheres were exposed to the hormone in the absence and presence of doxorubicin. The cell viability of CF41.Mg spheres exposed to a single dose of 1 mM melatonin decreased at 48 and 72 h (*p* < 0.0001), an effect that was equivalent to that induced by doxorubicin at 72 h. The drug combination did not show significant changes in cell viability in contrast with melatonin alone (Figure 3).

On the other hand, in spheres derived from REM134 cells, the cell viability did not present changes in response to both concentrations of melatonin (0.1–1 mM) and the combination with 1 nM doxorubicin, both at 48 and 72 h (Figure 4). Thus, these cells are resistant to melatonin and doxorubicin.

Mitoxantrone is a drug that is usually used as an adjuvant therapy in dogs with highly malignant mammary tumors. However, no data are showing whether canine mammary stem tumor cells are sensitive to it.

Thereby, the optimal concentration of mitoxantrone was determined in spheres of both cell lines, where four concentrations were tested, 1, 10, 100, and 1000 nM. The cell viability did not decrease in CF41.Mg spheres in response to the different concentrations studied. In REM134 spheres, after 48 h of incubation, the cell viability decreased only in response to 10 nM mitoxantrone (*p* < 0.05). However, at 72 h, no significant differences were observed in response to the different concentrations analyzed (Figure 5). Thus, both cell types exhibited resistance to mitoxantrone. For subsequent analyses, the lowest concentration (1 nM) was used.

Regarding the combined use of mitoxantrone and melatonin, both at 48 and 72 h of incubation, a decrease in the cell viability of CF41.Mg spheres were observed in response to 1 nM mitoxantrone plus 1 mM melatonin to control (*p* ≤ 0.0001) (Figure 6). This combination induced a significant reduction in cell viability to the mitoxantrone condition alone at 48 h, but not at 72 h. Thereby, melatonin was not able to reverse mitoxantrone resistance. On the other hand, REM134 spheres were resistant to both drugs, as shown in Figure 7.

In order to evaluate whether the effect of melatonin on CF41.Mg spheres were dependent on binding to its MT1 receptor, and a pharmacological blockade was carried out with luzindole, a drug that selectively antagonizes MT1 and MT2 [31]. As shown in Figure 8, 50 μM luzindole did not reverse the melatonin effect at any of the times studied. Luzindole alone exhibited an equivalent effect to melatonin at 48 h; however, this effect was not consistent since it was not appreciated at 72 h of incubation.

One of the mechanisms linked to drug resistance in CSC is the presence of multi-resistance molecules; therefore, we studied whether melatonin modulates the gene expression of *MDR1* and *ABCG2* in CF41.Mg spheres, cells that were sensitive to the hormone. Both genes have been linked to chemoresistance in mammary cancer in humans and dogs. A decrease in the gene expression of *ABCG2* and *MDR1* was observed in CF41.Mg spheres in response to 0.1 mM (*p* < 0.0001) and 1 mM (*p* < 0.001) melatonin (Figure 9).

## 4. Discussion

A small proportion of tumor cells with stemness characteristics within the tumor microenvironment may partially explain tumor progression. Cancer stem cells have a high capacity for developing drug resistance [11]. In this regard, it is necessary to develop selective targeted therapies against this type of cells, to achieve better therapeutical control of mammary carcinomas in dogs. Melatonin has been identified as a safe and effective agent against many types of cancer, especially mammary neoplasms, interfering with tumor progression due to its potential to modulate cell proliferation and invasion, apoptosis, chronic inflammation, and angiogenesis [27]. Although its potential effects on mammary CSCs have been little studied, melatonin has exhibited some effects on this type of cells [29,30] that suggest that it could be a pharmacological alternative to inhibit them.

Mammary CSCs studied here showed the ability to form spheres in vitro, chemoresistance, and the expression of the CD44^+^/CD24^−/low^ phenotype, characteristics of CSCs. However, other properties are typical of CSCs such as radio-resistance, self-renewal capacity, tumor-initiating capacity, invasiveness, tumor cell differentiation, and the expression of various molecules that support stemness such as OCT4, SOX2, and Nanog, among others [13,14,18]. Some CSC features such as the tumor-initiating capacity, and high OCT4, and SOX2 expression have been associated with malignancy and a worse prognosis in canine mammary tumors [11,32].

It has been described that MT1 expression is predominant in breast tumor cells, which suggests a sensitivity to the cytotoxic activity of melatonin [33]. In this regard, CF41.Mg and REM 134 spheres expressed the *MT1* gene, but no *MT2* gene expression was observed. It has been reported that MT1 is expressed predominantly in estradiol-dependent canine mammary tumors, and, to a lesser extent, in estradiol-independent tumors, describing a positive correlation between the expression of MT1 and estradiol receptor alpha (ERα) [20]. The above is also valid for human breast tumors [21]; however, at the cellular level, MT1 expression also occurs in ER-negative breast tumor cells [34]. In this regard, a histological co-expression of MT1 with nestin—a stem cell marker—has been described in human breast tumor tissue [35], which implies that MT1 would also be expressed and would be relevant in highly malignant neoplasms that are negatively correlated with ERα. To date, we do not know if the cells studied here express a functional ER.

We and others have described that melatonin can negatively modulate mammary CSC, reducing its viability, invasion, and forming-spheres ability in vitro [29,30]; however, the phenotypic and functional characteristics of canine mammary CSC in the presence of the hormone have not yet been studied. Our outcomes show that the hormone does not induce a change in the proportion of cells expressing the CD44^+^/CD24^−/low^ phenotype, but it does reduce the cell viability depending on concentration and time. These effects were also dependent on the cell type, since REM134 spheres were resistant to its effect, as well as doxorubicin and mitoxantrone. To date, there are no precise data regarding the transcriptomic, epigenetic, or protein profile of the tumor cells studied here. Possible differences in pathways linked to the pharmacological action of the hormone could explain the different sensitivity observed. Elucidating this issue could have a strong impact on the identification of molecular targets for the potential therapeutic use of melatonin in mammary tumors in dogs.

As described and verified in this study, spheres of both cell types exhibited different stemness characteristics [13,14]. These CSCs express OCT4, a transcription factor that induces pluripotency in neural cells, and that could be a carcinogenic target in stem cells. Furthermore, OCT4 can promote radio-resistance by improving the process of the epithelial–mesenchymal transition, and tumor migration and invasion [36,37,38]. OCT4 can be downmodulated by melatonin in canine mammary spheres [29], which would explain the effect of the hormone on the viability of this type of cells. Moreover, it has been described in human breast cancer cells that melatonin reduced ER and OCT4 expression and the binding of the ER to OCT4, downregulating the sphere-forming ability, which reinforces that this hormone could modulate self-renewal in CSCs [39] mediated by an effect on OCT4. On the other hand, melatonin has preliminarily shown the ability to inhibit vasculogenic mimicry in vitro, an endothelial cell-independent pro-angiogenic mechanism linked to CSCs, that promotes tumor progression and resistance to antiangiogenic drugs [40,41].

Interestingly, the detrimental effect triggered by melatonin on the cell viability of CF41.Mg spheres was equivalent to that exerted by doxorubicin and significantly greater than that of mitoxantrone, which could have a positive practical impact given that the clinical use of the hormone in dogs is safe and does not affect general or blood health parameters [42].

By combining both chemo-drugs with the hormone, no additive effect was evident; therefore, melatonin did not reduce drug resistance in the concentrations and times analyzed. These results differ from those reported by other authors, where melatonin can reverse drug resistance [43,44]; however, some of these observations have been obtained with concentrations and evaluation times greater than those used in this study (2–4 mM, and 8–10 culture days, respectively). For example, melatonin 2–4 mM potentiates the cytotoxic effects of lapatinib and neratinib in HER-positive human breast cancer cells promoting stress in the endoplasmic-reticulum-induced unfolded protein response (UPR) and lysosomal degradation of the HER, respectively [43,44].

The effect of melatonin in CF41.Mg spheres described here would be independent of MT1, since, when pharmacologically blocking this receptor with luzindole, no loss of the melatonin effect was observed. Thus, since the hormone has a lipophilic nature, it could interact with intracellular Ca^2+^-regulatory protein calmodulin (CaM), leading to a decreased sensitivity of adenylate cyclase (AC) in binding to CaM [21]. The lower activity of AC induces a reduction in the intracellular concentration of cyclic adenosine monophosphate (cAMP) that alters protein kinase A (PKA), (cAMP) binding protein (CREB), and p300 coregulator expression/activation. Moreover, a decrease in the phospho-activation and transactivation of various transcription factors and nuclear receptors (NRs) including ERα [21] and orphan nuclear receptor ROR alpha (RZR/RORα) [26] will also occur. On the other hand, melatonin can downregulate telomerase activity [45], which should be elucidated in future mechanistic experiments.

The overexpression of several ABC transporters, including MDR1 and ABCG2, contributes to the multidrug resistance (MDR) phenotype, resulting in an efflux of drugs from tumor cells and decreased intracellular drug concentrations and toxicity. Normal and cancer stem cells show higher expression levels of several ABC transporters [16,46]. Previous studies have documented the widespread expression of various ABC transporters in canine mammary cancer, highlighting ABCG2 and MDR1, explaining the failure of chemotherapy [15,16,17,18,19]. Recently, Yang et al., 2023 [47] demonstrated the strong expression of MDR1 in a doxorubicin-resistant human breast cancer line (MCF-7/Adr). Therefore, to control MDR, we must understand how the expression of these transporters is regulated, to develop compounds that modulate their activity [48,49]. Melatonin inhibited in CF41.Mg spheres the gene expression of *MDR1* and *ABCG2*. These findings are consistent with what was described in previous studies, where the decrease in the expression and function of ABCG2 occurred after treatment with 1 mM melatonin in brain tumor stem cells and colon cancer by inducing the methylation of its promoter [50,51]. Hsieh et al., 2020 have reported similar results for the ABCB1 transporter in vincristine-resistant oral cancer cell lines, where melatonin inhibited the expression of the transporter, reduced its activity, and upregulated the susceptibility of resistant cells to apoptosis [52].

In our study, although melatonin negatively regulated *MDR1* and *ABCG2*, a decrease in chemoresistance was not observed, probably because this phenomenon requires a longer time to establish, which implies exposure times to the hormone greater than 72–96 h. As the mechanisms associated with drug resistance are multiple [15,16,17], it is also possible that many of them are activated and are not being modulated by the hormone, resulting in chemoresistance not changing in the presence of melatonin.

Some limitations of this study included not studying longer cell culture times of hormone exposure, to more accurately evaluate whether melatonin can modulate drug resistance. On the other hand, analyzing the protein expression and activity of MDR1 and ABCG2 in the presence of melatonin would have strengthened the findings described.

Since melatonin has shown pleiotropic anti-tumor effects on mammary tumor cells, it becomes necessary to analyze other pathways associated with drug resistance in CSCs to establish with greater certainty the conditions under which the hormone could modulate this phenomenon. However, our data suggest that melatonin is an interesting pharmacological candidate to study in clinical trials with female dogs with mammary carcinomas.

## 5. Conclusions

Melatonin modulates the survival of CSCs, an effect dependent on the concentration and cell type, and independent of MT1 binding. However, it does not exert an additive anti-tumor effect with cytotoxic drugs such as doxorubicin and mitoxantrone at the concentrations and times studied, despite downregulating the gene expression of *MDR1* and *ABCG2*.

## Figures and Tables

**Figure 1 animals-14-01229-f001:**
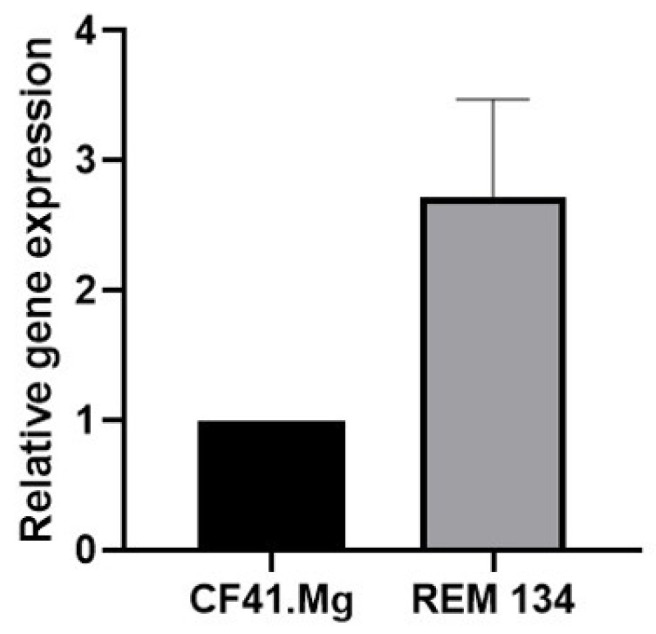
Gene expression of *MT1* in CF41.Mg and REM134 spheres. RT-qPCR assay. Values are means ± SD of 3 independent experiments.

**Figure 2 animals-14-01229-f002:**
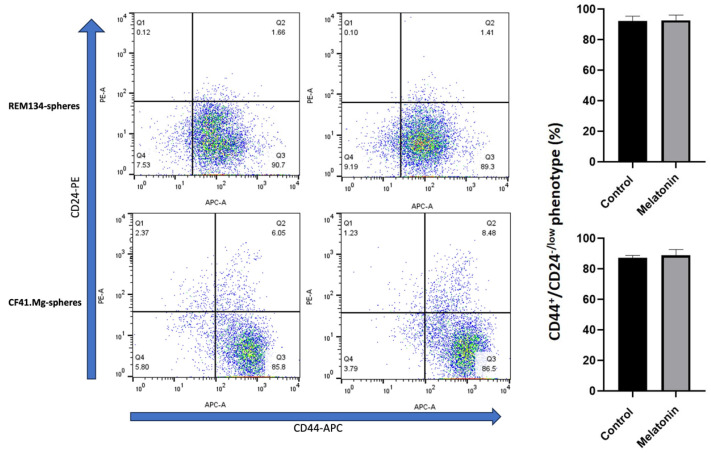
CD44^+^/CD24^−/low^ phenotype in REM134 and CF41.Mg spheres exposed to melatonin (1 mM). Flow cytometry assay. Representative plots of 3 independent experiments (**left panel**); quantification of % of cells exhibiting this phenotype (**right panel**), *p* > 0.05 (*T*-test).

**Figure 3 animals-14-01229-f003:**
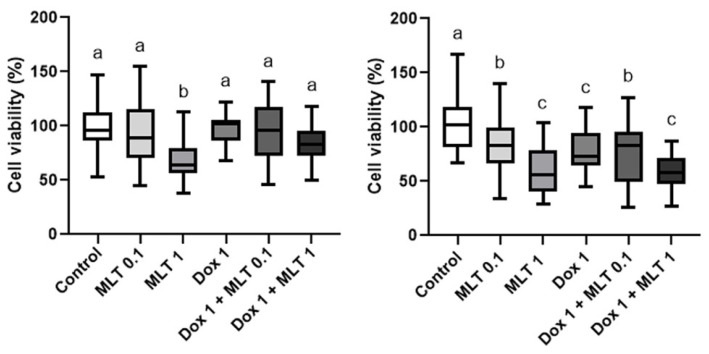
Cell viability of CF41.Mg spheres exposed to 1 nM doxorubicin (Dox) and different concentrations of melatonin (0.1–1 mM) (MLT) for 48 (**left**) and 72 h (**right**). MTS reduction assay. Values are means ± SD of 3 independent experiments carried out in triplicate. Different letters indicate statistical significance, *p* < 0.0001 (ANOVA and Bonferroni test).

**Figure 4 animals-14-01229-f004:**
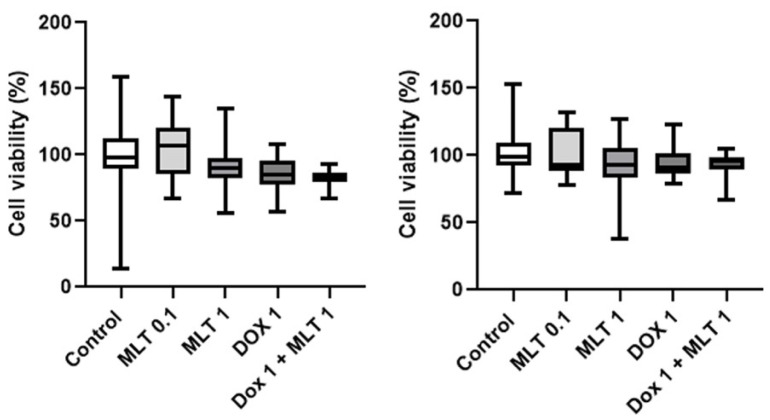
Cell viability of REM134 spheres exposed to 1 nM doxorubicin and different concentrations of melatonin (0.1–1 mM) for 48 (**left**) and 72 h (**right**). MTS reduction assay. Values are means ± SD of 3 independent experiments carried out in triplicate, *p* > 0.05 (ANOVA and Bonferroni test).

**Figure 5 animals-14-01229-f005:**
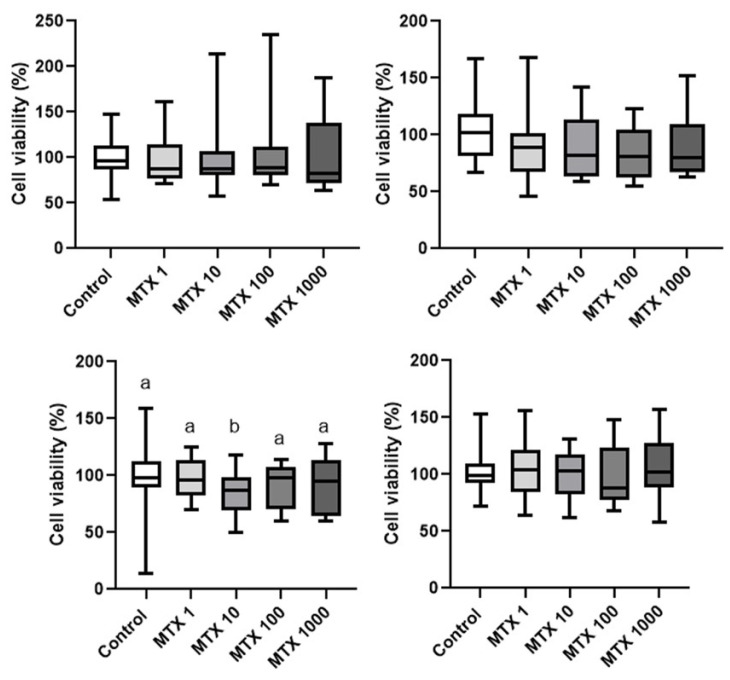
Viability of CF41.Mg (**upper panel**) and REM134 (**lower panel**) spheres in response to different concentrations of mitoxantrone (1–1000 nM) (MTX) for 48 (**left**) and 72 (**right**) h. MTS reduction assay. Values are means ± SD of 3 independent experiments carried out in triplicate. Different letters indicate statistical significance, *p* < 0.05 (ANOVA and Bonferroni test).

**Figure 6 animals-14-01229-f006:**
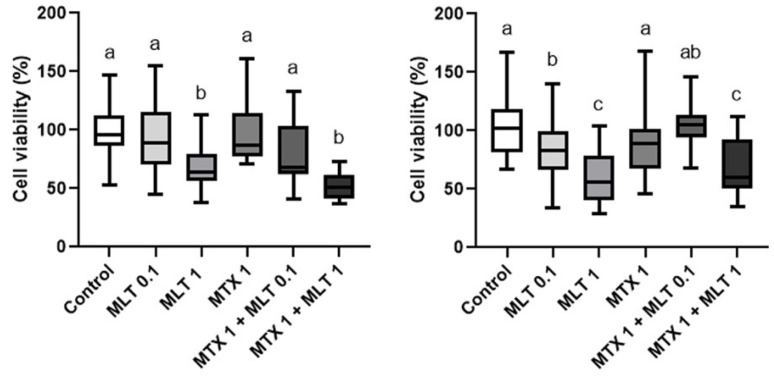
Cell viability of CF41.Mg spheres exposed to 1 nM mitoxantrone and different concentrations of melatonin (0.1–1 mM) for 48 (**left**) and 72 h (**right**). MTS reduction assay. Values are means ± SD of 3 independent experiments carried out in triplicate. Different letters indicate statistical significance, *p* < 0.0001 (ANOVA and Bonferroni test).

**Figure 7 animals-14-01229-f007:**
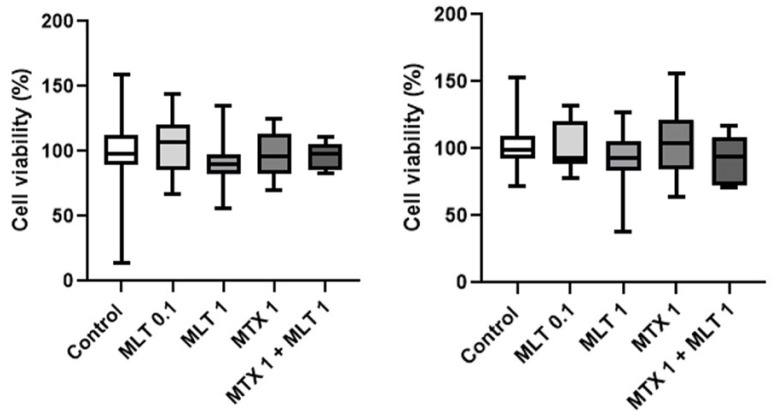
Cell viability of REM134 spheres exposed to 1 nM mitoxantrone and different concentrations of melatonin (0.1–1 mM) for 48 (**left**) and 72 h (**right**). MTS reduction assay. Values are means ± SD of 3 independent experiments carried out in triplicate, *p* > 0.05 (ANOVA and Bonferroni test).

**Figure 8 animals-14-01229-f008:**
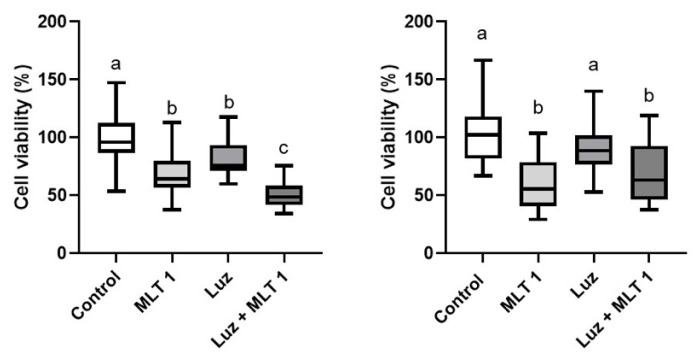
Viability of CF41.Mg spheres in response to melatonin 1 mM (MLT) and luzindole 50 μM (Luz) for 48 (**left**) and 72 (**right**) h. MTS reduction assay. Values are means ± SD of 3 independent experiments carried out in triplicate. Different letters indicate statistical significance, *p* < 0.05 (ANOVA and Bonferroni test).

**Figure 9 animals-14-01229-f009:**
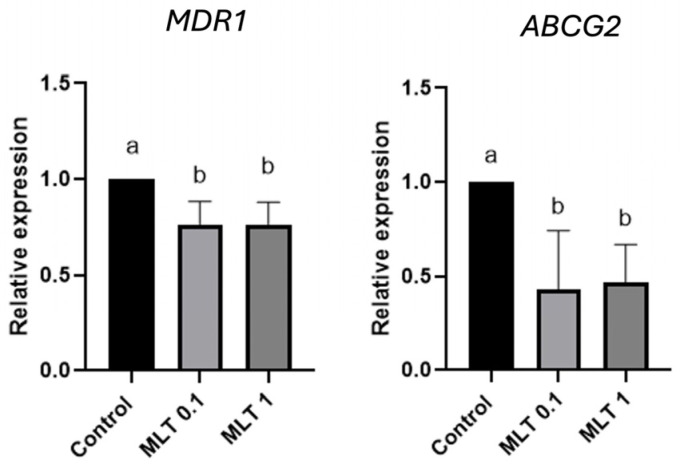
Gene expression of *MDR1* (**left** panel) and *ABCG2* (**right** panel) in CF41.Mg spheres in response to melatonin (0.1–1 mM). RT-qPCR assays. Values are means ± SD of 3 independent experiments. Different letters indicate statistical significance, *p* < 0.0001 (ANOVA and Bonferroni test).

**Table 1 animals-14-01229-t001:** Primers used for gene expression analysis through qRT-PCR (NR: not reported).

Gene	Genbank Access Nº	Primer Pair Sequences
*β actin*	NC_000071.7	FWD: 5′-CAAATGTGGATCAGCAAGCAG-3′ REV: 5′-GAAAGGGTGTAACGCAACTAAAG-3′
*MDR1*	NC_006596.2	FWD: 5′-ACAGGAGATTGGCTGGTTTG-3′ REV: 5′-AAGTCCAAGAACAGGGCTGA-3′
*ABCG2*	NC_006614.2	FWD: 5′-GACCTCCAACGACCTGAAGA-3′ REV: 5′-GAAGATTTGCCTCCACCTGT-3′
*MT1*	NR	FWD: 5′-TGTGCTTTCTAAACCTTTCTCCT-3′ REV: 5′- CACGAAGCCACTGATTTGGC-3′
*MT2*	NR	FWD: 5′-CTTGCTGACTTTTGCTCCCT-3′ REV: 5′-CGAATGACACTCAGCCCCAT-3′

## Data Availability

All the data are published in this manuscript or can be made available upon reasonable request.

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
