# Peer review of "Effect of Melatonin on Chemoresistance Exhibited by Spheres Derived from Canine Mammary Carcinoma Cells"

_animals, 2024, doi:10.3390/ani14081229_

Round 1
Reviewer 1 Report
Comments and Suggestions for Authors
"The background and discussion sections could be further improved, overall it feels average. I suggest delving deeper and providing more specific details in the discussion section."
Author Response
We appreciate the comments. We have improved the introduction and discussion, providing greater details, as suggested
Reviewer 2 Report
Comments and Suggestions for Authors
This Manuscript under Review proposes melatonin as an adjunctive lytic tool in treating canine mammary cancer. Authors astutely note that melatonin has shown anti-tumorogenic utility on mammary tumor cells; "however, its effects have been poorly evaluated in canine mammary CSCs". Therefore, authors sought to interrogate the effect of melatonin on the chemoresistance exhibited by spheres derived from canine mammary carcinoma to the cytotoxic drugs, doxorubicin and mitoxantrone. The presented results indicate that melatonin negatively modulated cell survival of CSCs derived from phenotypic CF41.Mg cells, which Authors describe as evidently independent of their MT1 receptor. As similar etiological bases may occur in both human and canine mammary cancer the investigation of novel means to arrest and reverse disease progression is of significant relevance to clinicians, patients & investigators.
Here, Reviewer finds fault not with methods employed but with Author's failure to address the fundamental underlying commonalities that bridge their proposed use of melatonin and its relevance in the specific cell species tested. As both canine and human mammary glands constitute ectodermal appendages, their lineage from neural ectodermal substrate seems to be a fundamental gap. Particularly given the modulatory capacity of melatonin in gene transcription and earliest gastrulation & cell differentiation a thread of functional continuity seems relevant.
Reviewer comprehends that the primary purpose of the proffered Manuscript may not be to answer the question "why is melatonin of use at all in this setting?". Still, if the overarching purpose of the Paper is to familiarize and educate clinicians and other interested Parties in melatonin as a potentially useful anti-neoplastic tool, the coherence of this fundamental bridge seems potentially helpful - if not glaringly axiomatic. Reviewer urges Authors to step back and take a slightly larger view of what they're actually aiming to communicate. That this remarkable molecule melatonin is implicated in a spectrum of fundamental effects from the earliest simple creatures thru to our own human species is shot thru the literature widely and in great depth.
Comments on the Quality of English LanguageMinor grammatical errors will benefit from a more careful Review by Authors.
Author Response
We appreciate the reviewer's comments. It is interesting what was raised regarding the link between melatonin and its study on canine mammary cancer stem cells (CSCs). We developed the link where these cells express OCT4, a transcription factor associated with stemness, which can be inhibited by melatonin. This already published information motivated us to analyze whether the hormone could modulate chemoresistance, one of the characteristics of CSCs.
On the other hand, in the discussion, we attempted to make a more general description regarding the potential impact that our results could have in the search for new therapeutic alternatives for mammary cancer in dogs.
Reviewer 3 Report
Comments and Suggestions for Authors
Spheres from the two canine mammary carcinoma cell lines CF41.Mg and REM 134 were used to study the in vitro effect of melatonin in combination with doxorubicin, mitoxantrone and luzindole. Methods used were flow cytometry, cell viability assay, qPCR and Western blot. Melatonin reduced the viability of CF41.Mg spheres in an independent way of its MT1 receptor. Drug combination did not show significant changes in cell viability. The REM cells did not respond to melatonin and doxorubicin.
Suggestion
The sphere formation and CD44+/CD24-/low phenotype was used as inclusion criteria for cancer stem cells (CSCs). The authors could discuss that there are further criteria for CSCs.
Minor comments
Page 2, line 70. It has …been…described.
Page 5, line 211. Reference missing.
Page 8, line 267. Reference missing.
Author Response
We appreciate all the comments made.
Other characteristics of CSC are incorporated into the discussion section (highlighted in yellow).
Minor comments are welcomed and corrected
Reviewer 4 Report
Comments and Suggestions for Authors
The manuscript by Cataldo et al. entitled Effect of melatonin on chemoresistance exhibited by spheres derived from canine mammary carcinoma cells aimed to evaluate the effect of melatonin on cancer stem cells from mammary gland tumor cell lines. Please see my specific comments below.
Introduction section line 64: “However, affected animals can develop drug-resistance, progressing to a fatal condition”. Mammary gland tumors (MGT) show low effects against chemotherapy. Usually, mammary gland tumors are naturally chemoresistant since they do not respond to chemotherapy as occurs in human breast cancer. The reference provided is about cancer in general not specifically MGT. The way the authors wrote the phrase induces the reader to think this is a common phenomenon in canine MGT.
The authors could provide a little background bout both cell lines. From where the cell lines were acquired? What type of cells?
How about the cell line certification and mycoplasma test?
Subheading 2.2 -> these cell lines were already tumorspheres? The authors did not provide any background about the sphere assay. Seems that flow cytometry was made on 2D cell lines, but the title and abstract induce the idea of the methodology based on tumorspheres.
Both antibodies were rat-anti-mouse. How about the cross-reactivity with the canine? Are the cells from dogs? I did not know these cells and the authors did not provide any background about them.
Line 193-197: The authors did not show data evaluated in the results section. This makes no sense. “Data not shown” experiments can be used in very specific ways to discuss some data. Why is the data not shown in the results instead of showing the data?
Authors could do a graphic representation of the flow cytometry data.
Line 210-211: Why provide a sentence previously published (reference is lacking) instead of showing your results? You treated the cells with doxorubicin and evaluated the chemoresistant genes. You can provide and state based on your own results that cells were resistant.
The authors mentioned the use of a specific mitoxantrone dosage based on the maximum plasma concentration in dogs.
Author Response
We appreciate all comments. We respond following the same order described by the reviewer.
- Thank you very much for the observation, with which we agree. The phrase is deleted and replaced by "affected animals usually develop drug resistance, decreasing treatment efficacy". Moreover, the reference is changed to a more specific one
- In the methods section, information regarding the cells used is incorporated (highlighted in red)
- The cells used were negative for mycoplasma. A specific PCR kit was used for this (this information is incorporated in methods)
- The method of growing spheres is clarified from line 119 of the manuscript
- The antibodies used in flow cytometry assays cross-react with CD44 and CD24 of canine origin, as has been demonstrated in 2 previous publications (Torres et al, Oncol Rep. 2015, 33: 2235-2244; Mishishita et al.
Res Vet Sci. 2011, 91(2): 254-260).
- We apologize for this omission. The pertinent figure is added (Figure 1).
- An additional graphical representation of the expression of the CD44+/CD24-/low phenotype in CF41.Mg and REM134 spheres is added (Figure 2).
- Thanks for the observation. We make a small clarification regarding the fact that the chemoresistance to doxo observed previously was also observed in this study.
- We apologize for this error. The sentence regarding the maximum concentration of mitroxanthone is deleted.
Round 2
Reviewer 1 Report
Comments and Suggestions for Authors
No other comments.
Reviewer 3 Report
Comments and Suggestions for Authors
The removal of the Western blot experiments is fine with me. I agree with that decision. Please note that the western blot in Figure 9 should be removed, not just in the legend .
Reviewer 4 Report
Comments and Suggestions for Authors
I have no further comments.
Round 3
Reviewer 4 Report
Comments and Suggestions for Authors
I have no further comments.